# Potyvirus HcPro Suppressor of RNA Silencing Induces PVY Superinfection Exclusion in a Strain-Specific Manner

**DOI:** 10.3390/ijms262311644

**Published:** 2025-12-01

**Authors:** Vincent N. Fondong, Prakash M. Niraula

**Affiliations:** Department of Biological Sciences, College of Agriculture, Science and Technology, Delaware State University, 1200 North DuPont Highway, Dover, DE 19901, USA; prakash.niraula@chase.com

**Keywords:** HcPro, potato virus Y, superinfection exclusion, suppression of RNA silencing, systemic acquired resistance (SAR)

## Abstract

The potyvirus helper component proteinase (HcPro) is a multifunctional protein, with one of its most documented functions being host antiviral RNA silencing suppression. This study shows that the HcPro of potato virus Y (PVY), an important member of the potyvirus group, prevents the replication of a related competing secondary virus. This phenomenon, referred to as superinfection exclusion (SIE), is common in bacterial, human, and plant virus infections. We also report that HcPro’s induction of SIE is strain-specific and that this specificity is provided by the first four amino acid residues of the protein. Consistent with the mechanism of SIE, the study found that HcPro does not exclude a resident virus. Additionally, HcPro’s induction of SIE was observed to function independently of its ability to suppress antiviral RNA silencing. HcPro’s induction of SIE is relevant given the prevalence of multiple PVY strains that routinely co-infect the same cell and that may lead to recombination and emergence of new and more virulent strains. Furthermore, cross-protection or systemic acquired resistance (SAR) that is employed in plant virus disease management occurs when SIE moves from the cellular level and spreads systemically, emphasizing the importance of studying SIE.

## 1. Introduction

Diseases caused by potato virus Y (PVY) are a major constraint to potato (*Solanum tuberosum* L.) production worldwide. PVY is disseminated in potatoes by the aphid vector and through potato seed tubers, which are the main mode of potato propagation. PVY-infected potato plants display leaf mosaic, crinkling, deformation, and leaf and tuber necrosis, all of which contribute to decreases in tuber yield and quality. Based on the host’s hypersensitive response, genome sequence, and/or serological properties, PVY has been grouped into three main clusters: PVY^O^, PVY^N^, and PVY^C^ [1]. Recombination amongst these clusters has resulted in new variants, and to date, at least nine recombination patterns of PVY^O^ and PVY^N^ sequences have been identified in potato-infecting PVY isolates, the most important being PVY^NTN^, PVY^N:O^, and PVY^N-Wi^ [2,3].

The PVY genome is a positive-sense single-stranded RNA [(+)ssRNA)] that encodes a large polyprotein of ~3062 amino acid residues. The coding region is flanked by 5′ and 3′ untranslated regions (UTRs), and a virus-encoded VPg (viral protein genome-linked) is covalently attached at the 5′ UTR, while the 3′ UTR is polyadenylated [4]. Upon translation, the polyprotein undergoes proteolytic processing, producing 10 functional proteins: protein 1 (P1), helper component-protease (HC-Pro), protein 3 (P3), 6 kilodalton (kDa) peptide 1 (6K1), cylindrical inclusion protein (CI), 6 kDa peptide 2 (6K2), viral protein genome-linked (VPg), nuclear inclusion-a protease (NIa-Pro), nuclear inclusion-b protein (NIb), and capsid protein (CP) [5]. Proteolytic processing is carried out by P1, HcPro, and NIa-Pro [6,7,8]. During replication, the slippage of the RNA-dependent RNA polymerase (RdRp) at the conserved motif (5′-GAAAAAA-3′) in the P3 cistron results in the expression of an additional protein named P3N-PIPO (N-terminal half of P3 fused to pretty interesting Potyvirus ORF) [9,10,11,12].

The potyvirus HcPro is a multifunctional protein (Figure 1), and one of the most studied potyviral proteins. Its most commonly documented functions are the ability to self-cleave from the viral polyprotein [13,14], and the suppression of RNA silencing [15,16,17,18,19]. Other HcPro functions include the mediation of aphid transmission of PVY [20], assembly of PVY particles [21,22], facilitation of local and systemic spread of progeny virus [23,24,25].

The ever-expanding functions of the potyviral HcPro include the ability to elicit superinfection exclusion (SIE), as was reported for the HcPro of potato virus A [27]. In SIE, a virus-infected cell is protected from subsequent infection (superinfection) by the same or a closely related virus [28,29,30]. Other potyviral proteins that have been reported to elicit SIE are NIa-Pro, CP, and P3 [31,32]. Virus-encoded protein inducers of SIE have been reported in other plant virus groups, including Orf1a, p33, and p23 of Citrus tristeza virus [30,33], and the matrix protein of Sonchus yellow net virus [34]. The induction of SIE has also been reported in human, animal, and especially bacterial viruses, including influenza A virus neuraminidase (NA) [35], bovine viral diarrhea virus E2 protein [36], and bacteriophage Pf PA0721 protein [37].

The phenomenon of cross-protection or systemic acquired resistance (SAR), where plants pre-infected with a mild isolate of a virus become protected against secondary infections by more severe isolates of the same virus, is used to manage plant virus diseases [38,39,40,41,42,43]. Mechanistically, it has been shown that SAR occurs when SIE moves from the cellular level and spreads systemically to the rest of the plant [38,44,45]. Therefore, to determine the establishment of SAR in PVY infections, we sought to characterize HcPro’s induction of PVY SIE using four PVY strains common in North America: PVY^NTN^, PVY^O^, PVY^N-Wi^, and PVY^N:O^ [46]. The results of the study show that the HcPro of PVY^O^ and of PVY^NTN^ causes the exclusion of the superinfecting parent virus strain, but not that of the other strain. The study further shows that this specific exclusion is provided mainly, but not exclusively, by the first four amino acid residues of the HcPro cistron. Using mutational analysis, we further show that HcPro’s induction of SIE functions independently of its ability to suppress antiviral RNA silencing. Furthermore, the PVY^NTN^ HcPro amino acid residue Asp-419 has been shown to be responsible for the ability of PVY^NTN^ to induce potato tuber necrosis ringspot disease (PTNRD) [47]. Therefore, we investigated whether this amino acid residue also influences HcPro’s suppression of RNA silencing. The results show that swapping Asp-419 and Glu-419 between the HcPro of PVY^O^ and PVY^NTN^ attenuates the RNA silencing suppression properties of both proteins.

## 2. Results

### 2.1. HcPro Induces SIE in a Sequence-Specific Manner

Here we investigated the ability of this protein to inhibit a superinfecting virus using the PVY^O^ and PVY^NTN^ strains and their encoded HcPro cistrons (designated here as HcPro^O^ and HcPro^NTN^, respectively). HcPro, expressed under the control of cauliflower mosaic virus 35S promoter, was agroinfiltrated to one side of the midrib of *N. tabacum* leaf blade and the other side was agroinfiltrated with an empty vector (negative control). After 2 days, PVY^O^ or PVY^NTN^ was mechanically inoculated to the whole leaf. At 7 days post virus inoculation, total RNA was extracted and viral RNA quantified using reverse transcription quantitative PCR (RT-qPCR). (In these experiments, the empty vector and the HcPro construct, which, in each case, were expressed on either side of the same leaf, were independent experiments from all other constructs).

The results from four independent experiments showed that tissues expressing HcPro^O^ displayed lower levels of PVY^O^ RNA than control tissues (Figure 2A). In contrast, HcPro^NTN^ was observed to cause an increase in levels of PVY^O^ RNA. Correspondingly, levels of PVY^NTN^ were significantly lower in tissues expressing HcPro^NTN^ (Figure 2B), while levels of PVY^O^ were higher, compared with the control samples. Thus, HcPro inhibits the cognate virus strain while enhancing levels of the other PVY strain. These results suggest that HcPro restricts the entry and/or replication of the cognate virus, i.e., it elicits the exclusion of the superinfecting cognate virus.

PVY^N-Wi^ and PVY^N:O^ belong to the same PVY^N^ cluster as PVY^NTN^, and their encoded HcPro cistrons are within the same PVY^N^ recombinant fragment with only one amino acid substitution between the HcPro^NTN^ and HcPro of PVY^N-Wi^ (K272N) and that of PVY^N:O^ (A48S) (Appendix A). We therefore reasoned that HcPro^NTN^, but not HcPro^O^, would likely elicit the superinfection of PVY^N-Wi^ and PVY^N:O^, based on the specificity of SIE. To investigate, HcPro^O^ and HcPro^NTN^ were agroinfiltrated into *N. tabacum*, followed by the inoculation of PVY^N-Wi^ and PVY^N:O^ as described above. Viral RNA quantification showed that tissues expressing HcPro^NTN^ recorded lower levels of PVY^N-Wi^ and PVY^N:O^ RNA than control tissues (Figure 2C,D). Accordingly, tissues expressing HcPro^O^ displayed lower levels of PVY^N:O^ RNA, suggesting that the sequence elements with which HcPro interacts to induce SIE are conserved between PVY^O^ and PVY^N:O^. In contrast, tissues expressing HcPro^O^ showed significantly higher levels of PVY^N-Wi^ (Figure 2D). These results are consistent with the specificity of the SIE mechanism. We note that HcPro^NTN^, HcPro^N-Wi^, and HcPro^N:O^, which all have the ^1^GVMD^4^ motif, show 36 amino acid differences with HcPro^O^ (Appendix A). Together, these data show that PVY deploys HcPro to regulate levels of the homologous PVY strain genomes in a sequence-specific manner.

### 2.2. Four N-Terminal Amino Acid Residues Provide Specificity of HcPro Induction of SIE

#### 2.2.1. Characterization of HcPro Induction of PVY^O^ SIE

Given that HcPro^O^ and HcPro^NTN^ restrict the homologous strain while enhancing levels of the other strain, we reasoned that there are amino acid residues or motifs that cause this dichotomy in the behavior of HcPro toward the two PVY strains. Thus, we swapped amino acid residues or motifs, which are either highly conserved between the two PVY clusters (^1^GVMD^4^ (HcPro^NTN^)/^1^RVLN^4^ (HcPro^O^) and show considerable biochemical differences [^245^NRL^247^ (HcPro^NTN^)/^245^IRK^247^ (HcPro^O^)] or were previously reported to cause differences in PVY strain symptoms [E^419^ (HcPro^NTN^)/D^419^ (HcPro^O^)]. The constructs were expressed from pEarley100 and the SIE induction properties of these constructs were examined following the procedures described above. The results from four independent experiments showed that, similar to the results recorded above, tissues expressing HcPro^O^ recorded lower levels of PVY^O^, while HcPro^NTN^ had the opposite effect (Figure 3(Ai,Aii)). Like for HcPro^NTN^, tissues expressing HcPro^O(GVMD)^ (HcPro^O^ containing ^1^GVMD^4^ from HcPro^NTN^) were found to display enhanced levels of PVY^O^ RNA compared with tissues containing the empty vector (Figure 3(Aiii)). This outcome suggests that ^1^RVLN^4^ is involved in the HcPro^O^ induction of PVY^O^ SIE, and ^1^GVMD^4^ from HcPro^NTN^ likely mediates the HcPro^NTN^ enhancement of PVY^O^ levels. Furthermore, no significant differences in PVY^O^ RNA levels were observed between tissues expressing HcPro^NTN(RVLN)^ and the negative control (Figure 3(Aiv)), indicating that ^1^GVMD^4^ is involved in the ability of HcPro^NTN^ to enhance PVY^O^ levels. Additionally, unlike HcPro^NTN^, HcPro^NTN(IRK)^ did not significantly enhance PVY^O^ (Figure 3(Avi)), suggesting a role for the ^245^NRL^247^ motif in the ability of HcPro^NTN^ to enhance PVY^O^ levels. This indicates that, in addition to the first amino acid residues, other amino acid residues are involved in an effective HcPro induction of SIE. Correspondingly, tissues expressing HcPro^O(D419E)^ and HcPro^NTN(E419D)^ did not display significantly different levels of PVY^O^ (Figure 3(Avii,Aviii)), indicating that these amino acids are required for HcPro’s induction of SIE.

#### 2.2.2. HcPro^O^ Does Not Induce Exclusion of a Resident PVY^O^

In the experiments described above, HcPro was expressed prior to the inoculation of PVY^O^, which was therefore the superinfecting virus. To determine whether HcPro will have a similar influence on a resident or primary virus, HcPro cistrons and HcPro mutants were each agroinfiltrated on one side of the leaf, and an empty vector on the other side of leaf was systemically infected by PVY^O^ at 21 days post infection (dpi). Interestingly, unlike the resident HcPro^O^, the superinfecting HcPro^O^ did not inhibit PVY^O^, while HcPro^NTN^ and HcPro^O(GVMD)^ caused a decline in the level of PVY^O^ (Figure 3(Bi–Biii)). These differential observations between a primary and a secondary infection are consistent with the mechanism of SIE, which only occurs on a superinfecting virus [28,29,30]. Further analyses showed that, unlike the resident HcPro^NTN(RVLN/IRK)^, the superinfecting HcPro^NTN(RVLN/IRK)^ caused an increase in the levels of PVY^O^ (Figure 3(Bv)). Taken together, these data show that HcPro^O^ has evolved to regulate the superinfecting cognate PVY^O^, but not a resident virus.

#### 2.2.3. HcPro^NTN^ Induction of PVY^NTN^ SIE

We also characterized the induction of PVY^NTN^ SIE in leaf tissues expressing mutants of HcPro^NTN^ and HcPro^O^. Here, similar to the observations made above (Figure 2A), tissues expressing HcPro^NTN^, but not those expressing HcPro^O^, displayed lower levels of PVY^NTN^ RNA compared with control tissues (Figure 4(Ai,Aii)). However, unlike HcPro^NTN^, tissues expressing HcPro^NTN(RVLN)^ did not display lower levels of PVY^NTN^ RNA, indicating that ^1^GVMD^2^ plays the same role in the HcPro^NTN^ induction of PVY^NTN^ SIE, as does ^1^RVLN^2^ in the HcPro^O^ induction of PVY^O^ SIE. Strikingly, the replacement of ^245^NRL^247^ with ^245^IRK^247^ from HcPro^O^ reconstituted the ability of HcPro^NTN(RVLN)^ to induce PVY^NTN^ SIE, as evidenced by the fact that HcPro^NTN(RVLN/IRK)^ caused a substantial reduction in the levels of a superinfecting PVY^NTN^ RNA (Figure 4(Av)). The role of ^245^IRK^247^ in the reconstitution of SIE is confirmed by the fact that HcPro^NTN(IRK)^ also induced the SIE of PVY^NTN^ (4(Avi)). Furthermore, the mutants HcPro^NTN(D419E)^ and HcPro^NTN(E419D)^ showed no exclusion of PVY^NTN^, thus confirming a role for Asp-419 and Glu-419, respectively, in HcPro’s induction of SIE.

#### 2.2.4. Superinfecting HcPro Does Not Induce Exclusion of a Resident PVY^NTN^

We also investigated whether HcPro^NTN^ inhibits a resident PVY^NTN^. Like for the resident PVY^O^ above, *N. tabacum* leaves systemically infected by PVY^NTN^ were infiltrated with HcPro constructs, and with an empty vector, as described above. An analysis of PVY^NTN^ RNA levels showed that, unlike observations made on a superinfecting PVY^NTN^, the levels of the resident PVY^NTN^ RNA were enhanced by a superinfecting HcPro^NTN^ (Figure 4(Bi,Bii)). Moreover, in contrast to the superinfecting PVY^NTN^ experiment shown in Figure 4A, HcPro^NTN(IRK)^ and HcPro^NTN(RVLN/IRK)^ were found to enhance the levels of a resident PVY^NTN^ RNA (Figure 4(Bv,Bvi)). Together, these observations emphasize that HcPro is a typical inducer of SIE and may deploy its other function of suppressor of RNA silencing to a resident PVY.

#### 2.2.5. Subcellular Localization Does Not Influence Behavior of HcPro Mutants

Subcellular localization is critical for protein function, as it determines its access to interaction partners and provides the cellular environment needed for interactions. To determine whether differences in the HcPro construct induction of SIE was due to subcellular localization, we investigated subcellular localization using confocal microscopy. These constructs were cloned into pEarleygate101, which contains a C-terminally fused yellow fluorescent protein (YFP), and transformed into *Agrobacterium tumifaciens* strain GV3101, cultures of which were infiltrated into *N. benthamiana* plant leaves. Confocal microscope imaging at 48 h post infiltration showed that HcPro localizes in the cytoplasm and nucleus. All mutants showed a similar pattern of distribution between the nucleus and cytoplasm as wild-type HcPro (Figure 5). These observations show that cellular localization did not influence SIE induction.

Together, these observations show that HcPro-mediated superinfection exclusion is strain-specific, and that the HcPro of a superinfecting PVY is unable to exclude a resident cognate PVY strain. In SIE, specificity is provided mainly, but not exclusively, by the first four amino acid residues of the HcPro cistron. The results also show that the HcPro^O^ exclusion of PVY^O^ is stronger than the HcPro^NTN^ exclusion of PVY^NTN^.

### 2.3. HcPro Induction of SIE and RNA Silencing Suppression Are Regulated Through Distinct Mechanisms

#### HcPro’s Suppression of PTGS

From this study, it is clear that HcPro elicits the exclusion of a superinfecting cognate PVY strain (*cis* inhibition). It also mediates increased levels of a different viral strain, likely through its canonical RNA silencing suppression function. SIE and the suppression of RNA silencing produce opposing outcomes since silencing suppressors cause increases in virus levels by countering the host antiviral defense. This contrasts with SIE, which limits levels of a superinfecting homologous virus. Thus, there is a dichotomy in both HcPro functions, which are likely mediated by different amino acid residues and/or motifs. To characterize the RNA silencing suppression properties of the HcPro mutants described above, we used the classic GFP silencing in *N. benthamiana* gfp-transgenic plant model [49]. First, we assessed the suppression of post-transcriptional gene silencing (PTGS) of RNA expressed from the 35S promoter. The HcPro constructs described above and GFP cloned into plasmid pEarley100 were transformed into *Agrobacterium*, as described above. Each construct was then co-infiltrated with GFP into *N. benthamiana* line 16c plants as described previously [50,51,52]; control plants were co-infiltrated with an empty pEarley100 vector and GFP.

GFP fluorescence in infiltrated leaf patches was monitored with a handheld longwave UV lamp. The imaging results showed that all infiltrated *N. benthamiana* 16c leaf patches displayed bright green fluorescence within 2 days post infiltration (dpi), and by 5 dpi, there was a strong green fluorescence (Figure 6A). However, by 8 dpi, control leaves infiltrated with an empty vector and GFP had lost most of the green fluorescence, consistent with silencing of the *gfp* gene. In contrast, leaf patches expressing HcPro^O^ and HcPro^NTN^, as well as HcPro^O(GVMD)^ and HcPro^NTN(IRK)^, continued to display green fluorescence (Figure 6B), which persisted for up to 6 more days before being replaced by red chlorophyll fluorescence.

GFP levels were quantified using anti-GFP antibody (α-GFP). Here, total plant protein was extracted at 5 and at 12 dpi from infiltrated leaves and GFP levels determined in an immunoblot analysis. This analysis showed that at 5 dpi, there were similar levels of GFP in all infiltrated leaves (Figure 7). However, at 12 dpi, there were lower levels of GFP in samples expressing HcPro^NTN(RVLN)^, HcPro^NTN(RVLN/IRK)^, HcPro^O(D419E)^, HcPro^NTN(E419D)^, and pEarley100 vector, while samples expressing HcPro^O(GVMD)^ and HcPro^NTN(IRK)^ showed similarly high levels of GFP as tissues expressing wild-type HcPro^O^ and HcPro^NTN^. These observations indicate that swapping “^1^RVLN^4^” from HcPro^O^ to HcPro^NTN^ negatively affects the ability of HcPro^NTN^ to suppress RNA silencing; in contrast, “^1^GVMD^4^” appears not to affect HcPro^O^ suppression of silencing.

The pEarley101 plasmid to which HcPro constructs were cloned for subcellular localization contains a C-terminal HA tag fusion in addition to YFP. This allowed us to confirm that these HcPro constructs were expressed by carrying out an immunoblot analysis using anti-HA tag antibody (α-HA). All HcPro constructs were detected (Figure 7), confirming expression. In these immunoblot assays, the heat shock complex 70 (HSC70) housekeeping protein was the loading control.

## 3. Discussion

This study has shown that one of the PVY HcPro cistron multiple functions in the virus infection cycle is the induction of superinfection exclusion (SIE). SIE is a mechanism in which a resident or primary virus prevents homologous superinfecting viruses from entry into the cell [38,53]. Mounting evidence indicates that during co-infection by related viruses, including potyvirus strains, the co-infecting viruses largely mutually exclude and colonize separate cell clusters [54,55,56]. HcPro likely contributes to this segregation by excluding the superinfecting cognate virus. HcPro’s induction of SIE is particularly important because of the existence of multiple prevailing PVY strains, which routinely co-infect the same plant, a phenomenon that is common in viral infections [55,57,58]. It has been suggested that SIE is a winning strategy for viruses in the short term because it protects a cell from being infected by competing viruses, giving the resident virus a greater chance of proliferating [59]. However, excluding related virus strains promotes homogeneity in virus populations, and this prevents recombination and promotes the emergence of beneficial mutations that produce new adaptations, leading to a more fit population in the long term. Therefore, the HcPro induction of SIE likely influences virus population structure and, thus, has an impact on virus evolution.

Unsurprisingly, the inhibition of secondary infection by the primary virus has been suggested to be “collateral damage” from virus autoregulation [38], where the SIE mechanism is unable to distinguish between own progeny genomes and homologous superinfecting virus genomes. Naturally, viruses must limit exponential replication where progeny genomes replicate unchecked. To do this, the parent virus must develop a mechanism to prevent progeny genomes from replicating and overwhelming the system. Hence, these large progeny virus populations are the intended target of SIE, while the few homologous superinfectors are only an unintended target. This appears to be the mechanism exhibited by HcPro given that, in this study, HcPro^O^ and HcPro^NTN^ were observed to limit the levels of the cognate PVY strain and not those of the other strain. These observations partly explain why unrelated viruses generally interact in a synergistic manner, which has a facilitative effect on one or both viral partners, while related viruses antagonize [55,60]. It is not surprising that a superinfecting HcPro does not elicit SIE since the resident virus already autoregulates. Therefore, the process is stoichiometric, and “external” HcPro does not add to the process.

Mechanistically, the phenomenon of SAR, which is common in DNA and RNA plant virus infections [61,62,63,64], has been shown to be a result of SIE becoming systemic (organismal) [38]. Thus, it is caused by the ability of the primary virus to use at least one of its proteins to prevent the proliferation of its progeny and closely related viruses throughout the plant. SAR has been known since 1929 [65], and it is used to control plant viruses by infecting plants with a mild strain as protection against future infection by a more severe strain [38,39,40,41]. In this case, the mild strain as the primary virus establishes throughout the plant and excludes subsequent entry and replication by a more virulent strain, thereby protecting the plant from a severe disease.

One of the key defenses plants deploy to counter virus infection is antiviral RNA silencing, where invading viral RNA is targeted and degraded by the host. To overcome this host defense, viruses typically encode at least one protein suppressor of RNA silencing, which enables the virus to replicate and spread. HcPro is one of three identified suppressors of RNA silencing encoded by potyviruses, the others being VPg [66] and NIb (Niraula et al., 2025; unpublished data). Thus, HcPro both induces SIE and suppresses RNA silencing. SIE and the suppression of RNA silencing produce opposing outcomes, since SIE limits replication while silencing suppressors cause increases in virus levels. HcPro therefore appears to function as a “master regulator” in the virus infection cycle.

To uncouple HcPro induction of SIE and RNA silencing suppression, we used mutational analysis and showed that the two mechanisms are mediated independently. Two of the mutants that illustrate this are HcPro^O(GVMD)^ and HcPro^NTN(IRK)^. Similar to wild-type HcPro, HcPro^O(GVMD)^, in which ^1^RVLN^4^ of PVY^O^ HcPro was replaced with ^1^GVMD^4^ from PVY^NTN^ HcPro, was found to suppress GFP in *N. benthamiana* line 16c plants. However, unlike HcPro^O^, it significantly enhanced levels of a superinfecting PVY^O^. This shows that these mechanisms can function independently. Also of interest is HcPro^NTN(IRK)^, which, like HcPro^O(GVMD)^, suppressed RNA silencing; however, it also elicited SIE of the cognate PVY^NTN^ strain while HcPro^O(GVMD)^ did not.

Finally, we note that SIE provides new opportunities to manage plant virus diseases by taking advantage of recent advances in biotechnological strategies for crop improvement [67,68,69]. For example, mosquitoes that were engineered to express nsP2, a SIE-inducing viral protein, were found to be resistant to subsequent infection by Sindbis virus (SINV) and chikungunya virus (CHIKV) [70]. Our study identified specific HcPro sequence elements that induce SIE. Beyond expressing a whole gene, virus resistant plants or insect vectors can be engineered to express these SIE sequence elements using biotechnological strategies in plant virus management. Correspondingly, prime editing (PE), the novel CRISPR-based genome editing method, can be used for targeted and precise modification of the genome [71]. This provides an excellent opportunity to precisely insert sequences that elicit SIE into plant or insect vectors.

## 4. Materials and Methods

### 4.1. Plasmid Construction and Inoculations

The HcPro cistrons investigated in this study were RT-PCR amplified from PVY strains PVY^NTN^ and PVY^O^ using the primers shown in Table 1. Amplification products were introduced into the entry vector pDONR/Zeo, and then shuttled into the binary vector pEarleygate100 or pEarleygate101 [72] following the Gateway^TM^ technology protocol (Life Technologies, Carlsbad, CA, USA). These plasmids were transformed into *A. tumefaciens* strain GV3101 for transient expression in tobacco. Mutations were introduced in pDONR/Zeo HcPro clones with a Q5 Site-Directed Mutagenesis Kit (New England Biolabs, Ipswich, MA, USA) using the primers listed in Table 1. HcPro mutants were then shuttled from the pDONR/Zeo entry clone into pEarleygate100 or pEarleygate101. These constructs were introduced into *A. tumefaciens* (strain GV3101) using the freeze–thaw method [73], and cultures were grown overnight at 30 °C. Agrobacterium suspensions were pelleted and resuspended in an agroinfiltration buffer (10 mM MES, pH 5.5; 10 mM MgSO4; 100 μM acetosyringone) at an optical density at 600 nm of 0.8.

PVY strains were mechanically inoculated to three- to four-week-old *N. tabacum* cv. Samsun plants. Virus inoculum was prepared by homogenizing systemically infected leaves of *N. tabacum* plants in a 0.01 M phosphate buffer, pH 7.0 at a dilution of 1/20 (*w*/*v*). The plants were placed in a room at 16 h light and 22 ± 3 °C temperature. In these experiments, each HcPro construct and the empty vector (negative control) were expressed on either side of the midrib of the same leaf and constituted an independent experiment from all other constructs.

### 4.2. Isolation of RNA and Reverse Transcription Quantitative PCR (RT-qPCR)

Total RNA was isolated from leaf samples using the RNeasy Mini kit (Qiagen, Germantown, MD, USA) with modifications. Here, approximately 200 mg of ground leaf and tuber tissues, respectively, were aliquoted for total RNA extraction using a leaf extraction buffer (5 M guanidine thiocyanate, 50 mM Tris-HCl pH 8, 10 mM EDTA, 2% N-lauroylsarcosine). The quality and quantity of RNA were determined using a NanoDrop 1000 Spectrophotometer (Thermo Fisher Scientific, Waltham, MA, USA) and agarose gel electrophoresis. The Invitrogen SuperScript III Platinum™ SYBR™ green one-step RT-qPCR Kit (Thermo Fisher Scientific, USA) was used to quantify viral RNA in 50 ng of total RNA on a Light Cycler 480 system (Roche Diagnostics, Basel, Switzerland). The 10 µL reaction mix contained 5 µL SYBR Green Supermix, 0.25 µL SuperScript III reverse transcriptase (RT)/Platinum^®^ Taq DNA polymerase enzyme mix, 0.4 µL 100 µM forward and reverse primers, and 2 µL of total RNA. The RT-qPCR conditions were as follows: cDNA synthesis at 50 °C for 15 min; qPCR cycling at 95 °C for 5 min, and 45 three-step cycles, each at 95 °C for 15 s, 60 °C for 30 s, and 72 °C for 10 s, followed by a final melting curve of 65–95 °C for 5 s.

### 4.3. Determination of Relative Viral RNA Levels

Here, we used elongation factor 1 (EF1) as the housekeeping gene. To compensate for potential variation between runs and plates, EF1 Ct values were normalized by subtracting the median Ct value from individual EF1 Ct values in each replicate. Normalized EF1 Ct values were then subtracted from individual target Ct values to obtain normalized target Ct values, which were used to calculate target ΔCt and ΔΔCt as we described recently [46]. In the calculation of relative viral RNA accumulation, we removed variation from background signals unrelated to viral RNA target by subjecting healthy tissues to the same amplification conditions as for each treatment [74], followed by the calculation of mean ΔCt and ΔΔCt of the background noise. These background ΔCt and ΔΔCt values were then subtracted from the experimental ΔCt and ΔΔCt values, respectively, the results of which were used to calculate individual 2^−ΔΔCT^ values [48] to obtain relative viral RNA accumulation. Error bars were computed to indicate the standard deviation of three different technical repeats from three biological replicates; significant differences between 2^−ΔΔCT^ values were determined using a *t*-test, and differences were considered statistically significant when the *p*-value was less than 0.05.

### 4.4. Western Blot Analysis

At 5 and 12 dpi, leaves co-infiltrated with GFP and the respective HcPro constructs were extracted in a protein extraction buffer [51] and the proteins separated in a 4–12% SDS-PAGE reducing gel (Genscript, Piscatway, NJ, USA). The proteins were electrophoretically blotted to PVDF membranes, and then blocked in 5% milk in TBST for 1 h followed by incubation at 4 °C overnight with rocking in rabbit anti-GFP primary antibody (GenScript, USA) or rabbit anti-HA tag primary antibody (Proteintech, Rosemont, IL, USA), each at a dilution of 1:5000. Membranes were washed three times (5 min each) with TBST and then incubated with goat anti-rabbit IgG (H + L) secondary antibody, HRP (Invitrogen, Waltham, MA, USA), at a dilution of 1:1000, followed by three 5-minute washes with TBST. Chemiluminescent signals were generated using SuperSignal™ West Atto Ultimate Sensitivity Substrate (Thermo Fisher Scientific, USA), and emitted light was captured using a G:Box and GeneSys software version: 1.8.11 (Syngene, Iselin, NJ, USA).

### 4.5. Confocal Microscopy

Leaf pieces were excised, gently vacuum infiltrated with water, placed on a slide, and covered with a coverslip for confocal imaging. Images were acquired on an upright Zeiss LSM 780 laser-scanning microscope (Carl Zeiss, Inc., Oberkochen, Germany) using a Zeiss 40X C-Apochromat (NA 1.2) objective lens. Multichannel images of YFP and chloroplast autofluorescence were acquired using the 488 nm line of an Argon/Krypton laser with the 500 to 550 band pass and 650 long-pass 390/emission filters, respectively. Images were captured as single optical sections or as a z-series of optical sections, and z-series data sets were displayed as single maximum intensity projection generated with Zeiss Zen Black software vSP2.

## 5. Conclusions

The phenomenon of superinfection exclusion (SIE) plays a key role in virus infection dynamics and influences viral fitness, population diversity, and evolution. This is because excluding a related competing secondary virus from the cell that has been successfully infected by the primary virus eliminates competition for host resources, given that multiple infections allow for the exchange of genetic material between viruses through recombination and reassortment (for segmented genome viruses); this increases diversity and improves the efficiency of selection. However, this may decrease fitness by promoting the presence of deleterious mutants at low frequencies [59]. Our study shows that HcPro, a suppressor of RNA silencing, induces the SIE exclusion of closely related PVY strains. Thus, HcPro contributes to shaping the genetic structure of PVY and its evolution.

## Figures and Tables

**Figure 1 ijms-26-11644-f001:**
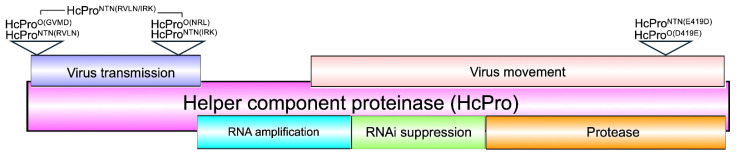
Functional domains of HcPro and mutations introduced in this study. The N-terminal region is involved in aphid vector binding and the transmission of the virus particle, while the central region participates in RNA silencing suppression, genome amplification, symptom severity, viral movement, and viral capsid-binding. The C-terminal domain displays proteolytic activity and self-cleaves from the viral polyprotein [26]. Open inverted triangles indicate amino acid residues that were mutated in this study.

**Figure 2 ijms-26-11644-f002:**
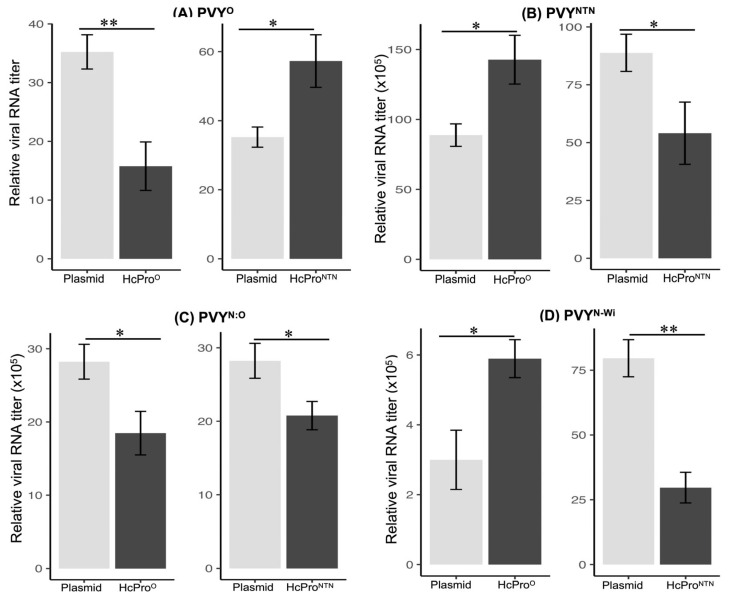
HcPro induces the exclusion of a superinfecting cognate PVY strain. HcPro of PVY^O^ (HcPro^O^) caused a decrease in PVY^O^ RNA levels while enhancing RNA levels of PVY^NTN^ and PVY^N-Wi^. In contrast, but consistent with the phenomenon of SIE, HcPro of PVY^NTN^ (HcPro^NTN^) induced a decrease in the RNA levels of members of the PVY^N^ cluster, PVY^NTN^, PVY^N-Wi^, and PVY^N:O^ and enhanced levels of PVY^O^. Relative RNA levels were determined using the 2^−ΔΔCT^ method [48]; 2^−ΔΔCT^ values represent the mean, and the error bars show the standard error of the means (of four independent experiments). A pair-wise *t*-test was used to determine the significance between 2^−ΔΔCT^ values; *, *p* < 0.05; **, *p* < 0.01. The underlying numerical data for this figure can be found in Appendix A.

**Figure 3 ijms-26-11644-f003:**
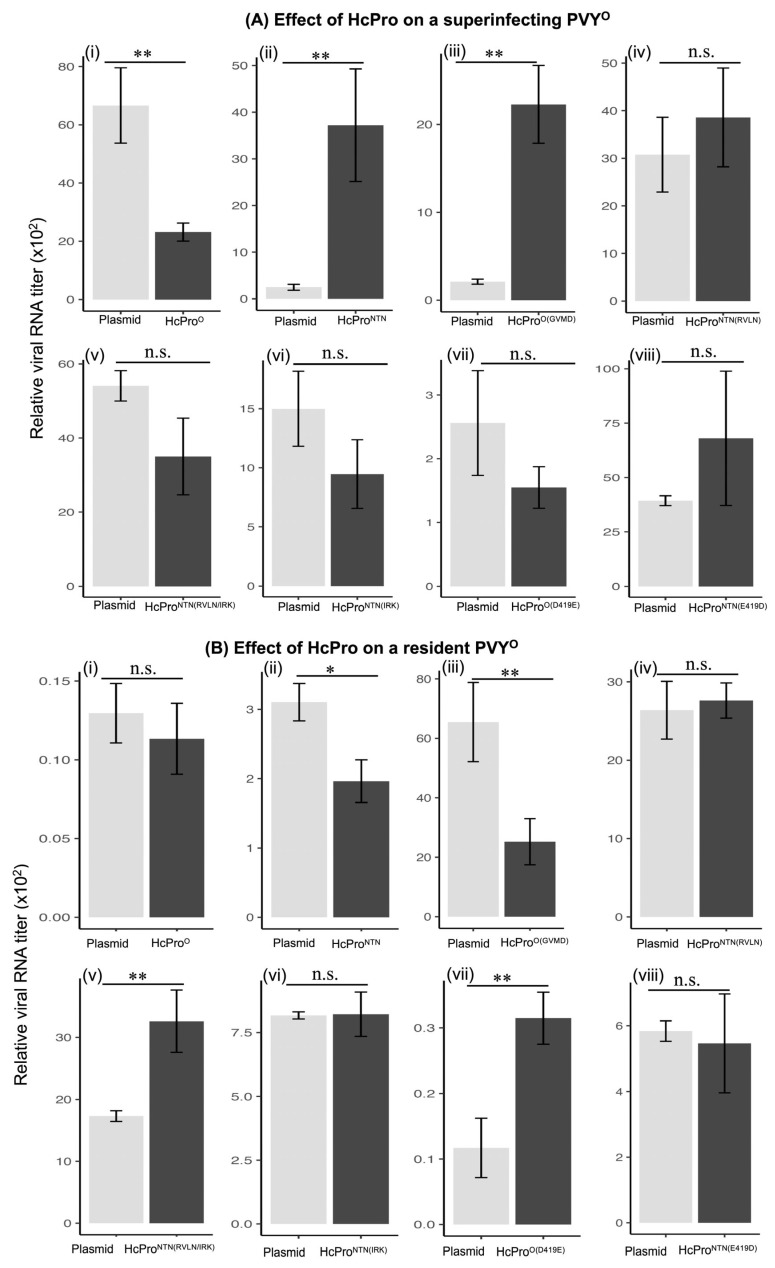
The N-terminal first four amino acid residues are specificity determinants of the HcPro-induced exclusion of a superinfecting, but not a resident, PVY^O^. (**A**) Effect of HcPro on a superinfecting PVY^O^: (i) HcPro^O^; (ii) HcPro^NTN^; (iii) HcPro^O(GVMD)^; (iv) HcPro^NTN(RVLN)^; (v) HcPro^NTN(RVLN/IRK)^; (vi) HcPro^NTN(IRK)^; (vii) HcPro^O(D419E)^; (viii) HcPro^NTN(E419D)^. Unlike HcPro^O^, HcPro^O(GVMD^ and HcPro^NTN^ caused an increase in levels of the superinfecting PVY^O^ RNA, while other mutants did not significantly affect PVY^O^ RNA levels. (**B**) Effect of HcPro on a resident PVY^O^: (i) HcPro^O^; (ii) HcPro^NTN^; (iii) HcPro^O(GVMD)^; (iv) HcPro^NTN(RVLN)^; (v) HcPro^NTN(RVLN/IRK)^; (vi) HcPro^NTN(IRK)^; (vii) HcPro^O(D419E)^; (viii) HcPro^NTN(E419D)^. Unlike the resident HcPro^O^, the superinfecting HcPro^O^ did not cause a significant decrease in the level of a resident PVY^O^ RNA, while HcPro^NTN^ and HcPro^O(GVMD)^ caused a decrease in the level of a resident PVY^O^ RNA. Relative RNA levels were determined using 2^−ΔΔCT^ as described above; 2^−ΔΔCT^ values represent the mean, and the error bars show the standard error of the means (of four independent experiments). A pair-wise *t*-test was used to determine the significance between 2^−ΔΔCT^ values; *, *p* < 0.05; **, *p* < 0.01; n.s., non-significant. The underlying numerical data for this figure can be found in Appendix A.

**Figure 4 ijms-26-11644-f004:**
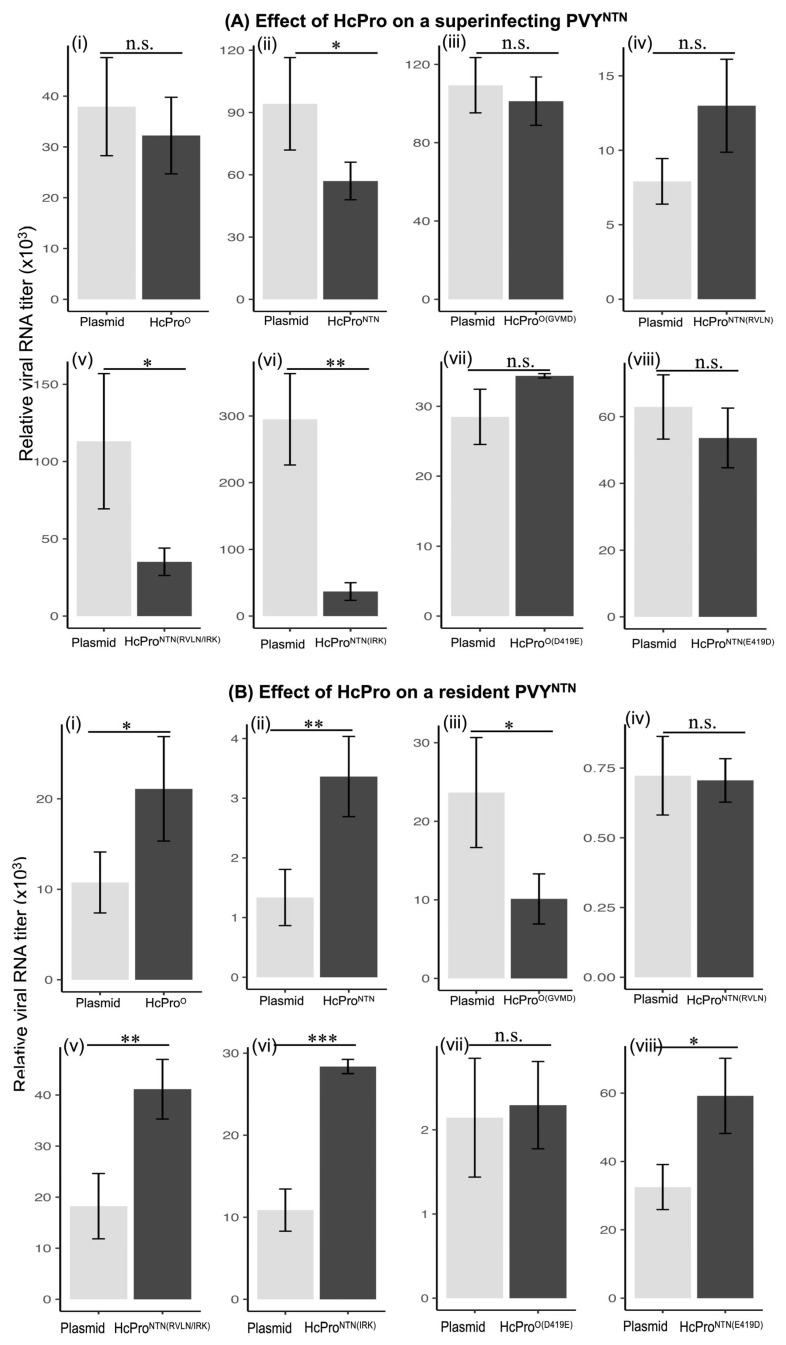
The N-terminal first four amino acid residues are specificity determinants of the HcPro^NTN^ induced exclusion of a superinfecting PVY^NTN.^ (**A**) Effect of HcPro on a superinfecting PVY^NTN^: (i) HcPro^O^; (ii) HcPro^NTN^; (iii) HcPro^O(GVMD)^; (iv) HcPro^NTN(RVLN)^; (v) HcPro^NTN(RVLN/IRK)^; (vi) HcPro^NTN(IRK)^; (vii) HcPro^O(D419E)^; (viii) HcPro^NTN(E419D)^. ^245^IRK^247^ from HcPro^O^ reestablished HcPro^NTN(RVLN)^ exclusion of a superinfecting PVY^NTN^, as indicated by HcPro^NTN(RVLN/IRK)^. (**B**) Effect of HcPro on a resident PVY^NTN^: (i) HcPro^O^; (ii) HcPro^NTN^; (iii) HcPro^O(GVMD)^; (iv) HcPro^NTN(RVLN)^; (v) HcPro^NTN(RVLN/IRK)^; (vi) HcPro^NTN(IRK)^; (vii) HcPro^O(D419E)^; (viii) HcPro^NTN(E419D)^. HcPro constructs tended to enhance levels of the resident PVY^NTN^. The 2^−ΔΔCT^ values represent the mean, and the error bars show the standard error of the means (of four independent experiments). A pair-wise t-test was used to determine the significance between 2^−ΔΔCT^ values; *, *p* < 0.05; **, *p* < 0.01; ***, *p* < 0.001; n.s., non-significant. The underlying numerical data for this figure can be found in Appendix A.

**Figure 5 ijms-26-11644-f005:**
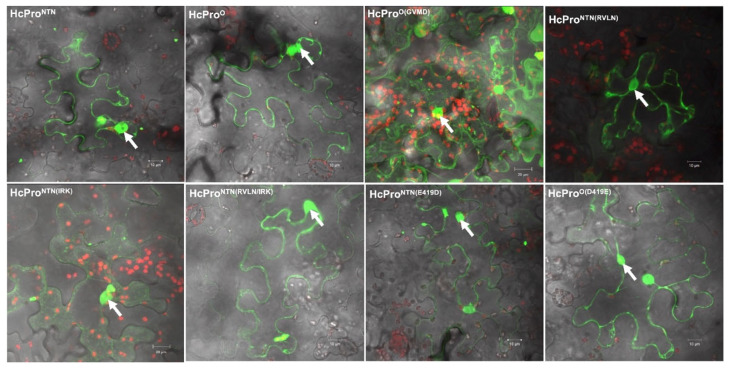
Subcellular localization of HcPro and mutants using confocal microscopy. Wild-type PVY^NTN^ and PVY^O^ HcPro and mutants HcPro^O(GVMD)^, HcPro^NTN(RVLN)^, HcPro^NTN(RVLN/IRK)^, HcPro^NTN(IRK)^, HcPro^O(D419E)^, and HcPro^NTN(E419D)^ all localized to both the nucleus (indicated by arrows) and the cytoplasm. Images were acquired on an upright Zeiss LSM 780 laser-scanning microscope (using a Zeiss 40× C-Apochromat (NA 1.2) objective lens.

**Figure 6 ijms-26-11644-f006:**
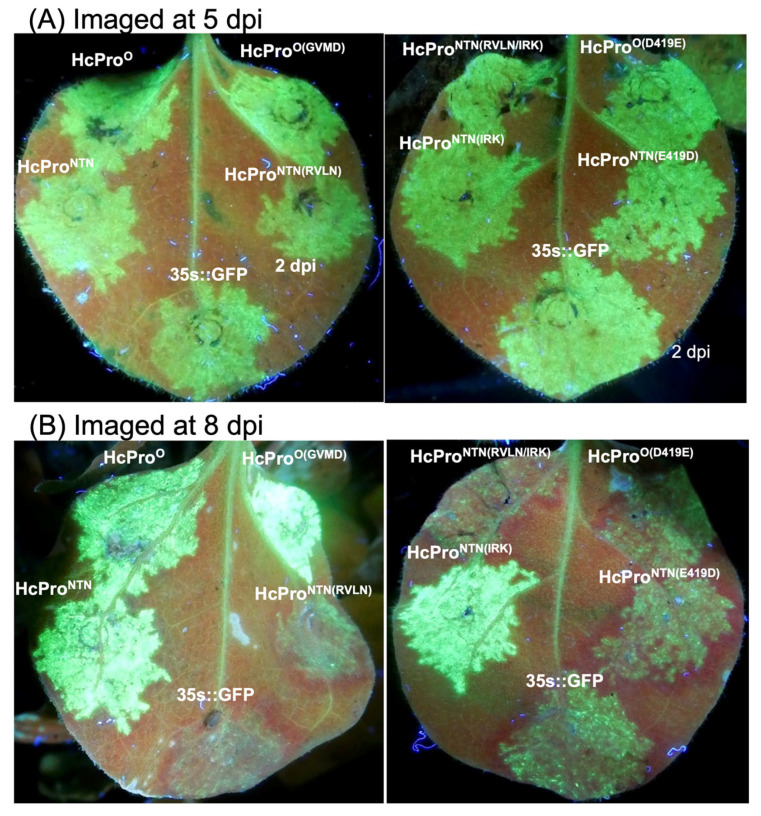
PVY HcPro suppression of post-transcriptional gene silencing (PTGS) in *N. benthamiana* line 16c GFP plants. *gfp* gene silencing suppression was assessed by co-infiltrating plants with GFP, and HcPro constructs. (**A**) At 5 dpi, all constructs displayed bright green fluorescence. (**B**) In contrast, at 8 dpi, HcPro^NTN(RVLN)^, HcPro^NTN(RVLN/IRK)^, HcPro^O(D419E)^, HcPro^NTN(E419D^, and GFP had lost much of the green fluorescence, while HcPro^NTN^, HcPro^O^, HcPro^O(GVMD)^, and HcPro^NTN(IRK)^ continued to display green fluorescence.

**Figure 7 ijms-26-11644-f007:**
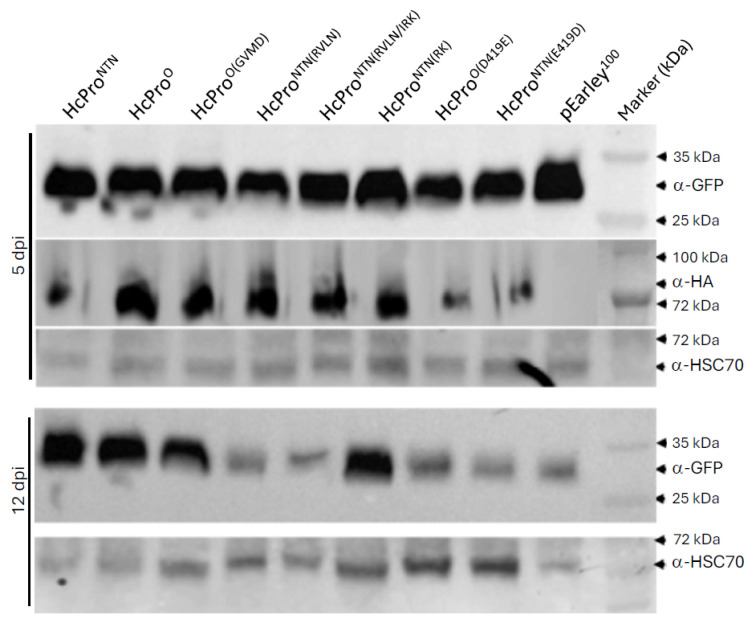
Western blot analysis of GFP in infiltrated 16c leaves infiltrated with HCPro^NTN^ and HcPro^O^, as well as constructs expressed from plasmid pEarleygate100. At 5 dpi, leaves co-infiltrated with GFP and HcPro^NTN^, HcPro^O^, HcPro^O(GVMD)^, HcPro^NTN(RVLN)^, HcPro^NTN(RVLN/IRK)^, HcPro^NTN(IRK)^, HcPro^O(D419E)^, HcPro^NTN(E419D)^, and pEarley100, respectively, were analyzed using anti-GFP antibodies. By 12 dpi, GFP showed lower levels in HcPro^NTN(RVLN)^, HcPro^NTN(RVLN/IRK)^, HcPro^O(D419E)^, HcPro^NTN(E419D)^, and pEarley100. For the loading control, membranes were stripped of the GFP antibody and probed with heat shock complex 70 antibody (α-HSC70).

**Table 1 ijms-26-11644-t001:** Primers used in this work.

Designation	Sequence	Construct
**NTN.Hc-Pro.F**	attB1 + GGGGTTATGGATTCA	HcPro^NTN^
**NTN.Hc-Pro.R**	attB2 + ACCAACTCTATAGTGCTTAATGTCAGACT	
**O.HcPro.F**	attB1 + CGTGTTTTGAACTCAA	HcPro^O^
**O.HcPro-Pro.R**	attB2 + ACCAACTCTATAATGTTTTATATCAGATTCTAATTCATCATTTGC	
**GVMD(1-4)RVLN-F**	TTGAACTCAATGGTTCAGTTCTCAAGC	HcPro^NTN(RVLN)^
**GVMD(1-4)RVLN-R**	AACACGGGCCATAAGGGCGAATTC	
**RVLN(1-4)GVMD-F**	ATGGATTCAATGATCCAGTTTTCGAATG	HcPro^O(GVMD^
**RVLN(1-4)GVMD-R**	AACCCCGGCCATTAAGCCTGCTTT	
**NRL(246-248)IRK-F**	CAAGCATCCGAATGGAACAAGAAAAC	HcPro^NTN(IRK)^
**NRL(246-248)IRK-R**	CGGATTTCATATGCTGAATAGCC	
**GVMD(1-4)RVLN-F**	TTGAACTCAATGGTTCAGTTCTCAAGC	HcPro^NTN(RVLN/IRL)^ *
**GVMD(1-4)RVLN-R**	AACACGGGCCATAAGGGCGAATTC	
**OHcProGlu419-F**	TTGACCATGAAACTCAAACGTG	HcPro^O(D419E)^
**OHcProGlu419-R**	CCAATATTCTAGGCAGTTC	
**NTNHcProAsp-F**	TCGATCACGACACGCAGACAT	HcPro^NTN(E419D)^
**NTNHcProAsp-R**	CTAGTATTCTAGGCAGTTCTG	

attB: site-specific recombination; * HcPro^NTN(RVLN/IRL)^ was obtained by using HcPro^NTN(IRK)^ primers and HcPro^NTN(RVLN)^ template.

## Data Availability

The raw data supporting the conclusions of this article will be made available by the authors on request.

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
