# Peer review of "Potyvirus HcPro Suppressor of RNA Silencing Induces PVY Superinfection Exclusion in a Strain-Specific Manner"

_ijms, 2025, doi:10.3390/ijms262311644_

Round 1
Reviewer 1 Report
Comments and Suggestions for Authors
This study showed that HcPro of PVYO and PVYNTN causes exclusion of the superinfecting parent virus strain but not that of the other strain, and that this specific exclusion is determined by the first four amino acid residues of the HCpro cistron. While the research is interesting and offers valuable data, several aspects related to data interpretation, figure presentation, and clarity of explanation need revision to improve the scientific rigor and overall readability of the manuscript.
- Figure 2 shows results from challenge inoculations with PVYO, PVYᴺᵀᴺ, PVYN:O, and PVYN-Wi. However, the figure legend only describes the results for PVYO and PVYᴺᵀᴺ. Please revise the legend to include explanations for all four challenge inoculations for clarity and completeness.
- Please verify and revise the statement “Viral RNA quantification showed that tissues expressing HcProNTN recorded lower levels of PVYN-Wi and PVYN: O RNA than control tissues (Figure 2B),” as Figure 2B presents results from PVYᴺᵀᴺ inoculation. The description appears inconsistent with the figure content and should be corrected accordingly.
- Figure 2C lacks a statistical analysis. Also, in Figure 2C, if the PVYN:O belongs to the same PVYN cluster as PVYNTN, then why does the HCProO elicit SIE of PVYN:O?
- The statement “We note that HcProNTN, HcProN-Wi, and HcProN:O show 36 amino acid differences with HcProO, including the first four amino acid residues 1RVLN2 and 1GVMD4” is ambiguous and could be misinterpreted as referring to eight differing amino acids. It would be clearer to specify that residues 1RVLN2 correspond to PVYO, while 1GVMD4 represents the N-type PVY strains examined in this study.
- Among the 36 amino acid differences identified, it is unclear why the authors selected specific residues for the swap analysis to identify amino acids involved in SIE induction. Could you please clarify the rationale or criteria used for selecting these particular residues?
- It would be helpful to include statistical significance markings directly in the figure panels (e.g., ** statistically significant, ns: non-significant), as it is hard to interpret statistical differences based only on the figure, legend, or text description.
- Statement:” Correspondingly, tissues expressing HcProO(D419E) and HcProNTN(E419D) did not display significantly different levels of PVYO (Figure 3A-vii & 3A-viii), indicating that these amino acids contribute to the HcPro induction of SIE” appears contradictory. If swapping the amino acid at position 419 does not affect PVY⁰ accumulation, it is unclear how this result supports the conclusion that these residues contribute to SIE induction. Please clarify or revise the interpretation.
- Figure 3 is titled “superinfecting PVYO and resident PVYO,” whereas the text refers to resident and superinfecting HcPro. Please clarify this discrepancy, as the differing terminology may lead to confusion.
- Statement:” Further analyses showed that, unlike the resident HcProNTN(RVLN/IRK), the superinfecting HcProNTN(RVLN) caused an increase in the levels of PVYO (Figure 3B-v)”, appears inconsistent with the data shown in Figure 3B-v, which only presents a comparison between the plasmid control and HcProNTN(RVLN/IRK). Please verify and correct the figure reference or description accordingly.
- Figure 3: The figure legend should concisely describe the key results presented in the figure rather than focusing on methodological details. Please revise the legend to emphasize the main findings.
- Figure 3: Both A and B panels have the same title,” A/ Effect of HcPro mutants on a resident PVYO.” “B/Effect of HcPro mutants on a resident PVYO.”
- Figure 4A: Each graph lacks a title or label, making it difficult to match the data with the corresponding explanations in the text. Please include clear graph names or headings to improve clarity and readability.
- Statement: “This further indicates that the N-terminal first four amino acid residues determine the specificity of SIE but are not the only amino acid residues involved in the process.” Can the author clarify what “determine the specificity of SIE” means here, since the results suggest that either wild-type HcProNTN or the mutant, which contains 1RVLN4 and 245IRK247 substitutions from HcProO, caused a substantial decline in the level of PVYNTN?
- Figure 6 does not contain labeled panels, yet the text refers to “Figure 6A” and “Figure 6B.” Please revise the figure to clearly differentiate and label the panels so that they correspond with the descriptions in the text.
- Statement: “Results confirmed that all HcPro were similarly expressed (Figure 7)”. is not fully supported by the data. Figure 7 shows noticeable variation in the intensity of the HcPro-HA bands at 10 dpi, whereas GFP appears consistently expressed across all samples. Please clarify this discrepancy and adjust the interpretation accordingly.
- Section 2.3.2 (“HcPro suppression of VIGS”): Although the authors mention minor differences between the VIGS and PTGS experiments, no further explanation or interpretation is provided regarding the significance of these differences. Please clarify the rationale for including the VIGS experiment and elaborate on how these results contribute to understanding the overall mechanism or function of HcPro in silencing suppression.
- M&M (Determination of relative viral RNA titer): Can the relative RNA levels from qPCR analysis of plant tissues (infected or healthy), as described in the text, be used to determine viral titer if a standard curve using viral plasmid DNA (or in vitro-transcribed RNA) is not included?, Please clarify whether the results represent relative viral RNA accumulation or absolute viral titer, and specify whether a standard curve was included in the analysis.
- M&M (Western blot analysis): The first paragraph appropriately describes the Western blot procedure. However, the second paragraph shifts abruptly to cellular observations. To improve clarity and organization, consider either placing this content in a separate section or revising the section subtitle to reflect both experimental components.
Comments on the Quality of English Language
I am not in a position to correct grammatical or language-related issues or to assess the quality of the English used in the manuscript.
Author Response
- Figure 2shows results from challenge inoculations with PVYO, PVYᴺᵀᴺ, PVYN:O, and PVYN-Wi. However, the figure legend only describes the results for PVYOand PVYᴺᵀᴺ. Please revise the legend to include explanations for all four challenge inoculations for clarity and completeness.
- PVYN:O, and PVYN-Wi have been added and the legend revised
- Please verify and revise the statement “Viral RNA quantification showed that tissues expressing HcProNTNrecorded lower levels of PVYN-Wiand PVYN: O RNA than control tissues (Figure 2B),” as Figure 2B presents results from PVYᴺᵀᴺ inoculation. The description appears inconsistent with the figure content and should be corrected accordingly.
- This has been corrected
- Figure 2C lacks a statistical analysis. Also, in Figure 2C, if the PVYN:O belongs to the same PVYNcluster as PVYNTN, then why does the HCProOelicit SIE of PVYN:O?
- We have indicated that sequence elements involved with HcPro induced SIE are common between the two strains
- The statement “We note that HcProNTN, HcProN-Wi, and HcProN:Oshow 36 amino acid differences with HcProO, including the first four amino acid residues 1RVLN2 and 1GVMD4” is ambiguous and could be misinterpreted as referring to eight differing amino acids. It would be clearer to specify that residues 1RVLN2 correspond to PVYO, while 1GVMD4 represents the N-type PVY strains examined in this study.
- This has been corrected
- Among the 36 amino acid differences identified, it is unclear why the authors selected specific residues for the swap analysis to identify amino acids involved in SIE induction. Could you please clarify the rationale or criteria used for selecting these particular residues?
- The reasons for choice have been included
- It would be helpful to include statistical significance markings directly in the figure panels (e.g., ** statistically significant, ns: non-significant), as it is hard to interpret statistical differences based only on the figure, legend, or text description.
- We have made this addition
- Statement:” Correspondingly, tissues expressing HcProO(D419E)and HcProNTN(E419D)did not display significantly different levels of PVYO (Figure 3A-vii & 3A-viii), indicating that these amino acids contribute to the HcPro induction of SIE” appears contradictory. If swapping the amino acid at position 419 does not affect PVY⁰ accumulation, it is unclear how this result supports the conclusion that these residues contribute to SIE induction. Please clarify or revise the interpretation.
- We have added that the amino acids are required for HcPro induction of SIE
- Figure 3 is titled “superinfecting PVYOand resident PVYO,” whereas the text refers to resident and superinfecting HcPro. Please clarify this discrepancy, as the differing terminology may lead to confusion.
- This has been clarified
- Statement:” Further analyses showed that, unlike the resident HcProNTN(RVLN/IRK), the superinfecting HcProNTN(RVLN)caused an increase in the levels of PVYO(Figure 3B-v)”, appears inconsistent with the data shown in Figure 3B-v, which only presents a comparison between the plasmid control and HcProNTN(RVLN/IRK). Please verify and correct the figure reference or description accordingly.
- This has been corrected
- Figure 3:The figure legend should concisely describe the key results presented in the figure rather than focusing on methodological details. Please revise the legend to emphasize the main findings.
- This has been done
- Figure 3:Both A and B panels have the same title,” A/ Effect of HcPro mutants on a resident PVYO.” “B/Effect of HcPro mutants on a resident PVYO.”
- This has been revised accordingly
- Figure 4A:Each graph lacks a title or label, making it difficult to match the data with the corresponding explanations in the text. Please include clear graph names or headings to improve clarity and readability.
- Titles have been added
- Statement: “This further indicates that the N-terminal first four amino acid residues determine the specificity of SIE but are not the only amino acid residues involved in the process.” Can the author clarify what “determine the specificity of SIE” means here, since the results suggest that either wild-type HcProNTNor the mutant, which contains 1RVLN4 and 245IRK247 substitutions from HcProO, caused a substantial decline in the level of PVYNTN?
- This revision has been made
- Figure 6does not contain labeled panels, yet the text refers to “Figure 6A” and “Figure 6B.” Please revise the figure to clearly differentiate and label the panels so that they correspond with the descriptions in the text.
- The labels have been added
- Statement: “Results confirmed that all HcPro were similarly expressed (Figure 7)”. is not fully supported by the data. Figure 7 shows noticeable variation in the intensity of the HcPro-HA bands at 10 dpi, whereas GFP appears consistently expressed across all samples. Please clarify this discrepancy and adjust the interpretation accordingly.
- New blots have been done and included
- Section 2.3.2 (“HcPro suppression of VIGS”):Although the authors mention minor differences between the VIGS and PTGS experiments, no further explanation or interpretation is provided regarding the significance of these differences. Please clarify the rationale for including the VIGS experiment and elaborate on how these results contribute to understanding the overall mechanism or function of HcPro in silencing suppression.
- VIGS section has been removed
- M&M (Determination of relative viral RNA titer):Can the relative RNA levels from qPCR analysis of plant tissues (infected or healthy), as described in the text, be used to determine viral titer if a standard curve using viral plasmid DNA (or in vitro-transcribed RNA) is not included?, Please clarify whether the results represent relative viral RNA accumulation or absolute viral titer, and specify whether a standard curve was included in the analysis.
- This has been changed to viral RNA
- M&M (Western blot analysis): The first paragraph appropriately describes the Western blot procedure. However, the second paragraph shifts abruptly to cellular observations. To improve clarity and organization, consider either placing this content in a separate section or revising the section subtitle to reflect both experimental components.
- The full description of Western blot has been included
Reviewer 2 Report
Comments and Suggestions for Authors
The manuscript “Potyvirus HcPro suppressor of RNA silencing induces PVY superinfection exclusion in a strain-specific manner” reports informative study addressing the role of HcPro in the mechanism of superinfection exclusion among plant viruses.
The study has scientific value and contributes to understanding the mechanisms of strain-specific superinfection exclusion in potyviruses. However, this work needs to be finalized and further clarified.
Major comments:
- Although confocal microscopy was used to visualize GFP fluorescence, the results are presented only qualitatively, based on visual inspection and descriptive terms such as “bright” or “weak” fluorescence. Such observations are inherently qualitative and subjective. Quantitative fluorescence measurements would make these data more objective and comparable.
- The discrepancies between the PTGS and VIGS results are not clearly explained. It was mentioned that some PTGS mutants lost activity, while in VIGS they retained it, but the authors do not discuss why. A possible biological interpretation of this observation is missing.
- Although the purpose of the study is to explore the relationship between SIE and silencing suppression, section 2.3.2 does not clearly show how the VIGS results relate to SIE.
- Compared to section 2.3.1 (PTGS), experiment 2.3.2 is described much more briefly, with fewer methodological details and limited interpretation.
- In section 2.3.1, HA-tag immunoblotting was performed to confirm HcPro expression, but this verification is missing for the VIGS experiment. It remains unclear whether all HcPro variants were expressed at comparable levels when delivered via the PVX vector.
- No quantitative correlation with the PTGS data is provided. The authors state that the results “generally coincide,” but no statistical comparison or correlation analysis is shown.
- The manuscript indicates that mutations (E419D, D419E, RVLN/IRK) affect VIGS efficiency, but does not discuss why these positions may be important, for instance, their potential structural or functional role in HcPro.
- The exact description of the PVX vector used in this work is not specified in the Methods section, which is important for reproducibility.
- Although the qRT-PCR data analysis is described in detail, the presentation of the results lacks transparency. The authors state that the data represent three biological replicates with three technical repeats; however, individual data points are not shown on the graphs, making it difficult to assess biological variability. It is strongly recommended to display individual replicates on the graphs and to include a supplementary table with raw or normalized ΔCt/ΔΔCt values to improve reproducibility and clarity.
- The Y-axes in different panels use inconsistent scales, which complicates visual comparison between experiments. Standardizing axis scales across all panels would facilitate a more accurate interpretation of relative changes.
Minor comments:
- The authors should consider adding information about the PVY strains used in the study to the abstract.
- The rationale for selecting these particular PVY strains should be clarified and justified in the manuscript.
- The quality and resolution of the figures 6 and 7 should be improved to enhance data visibility.
Author Response
- Although confocal microscopy was used to visualize GFP fluorescence, the results are presented only qualitatively, based on visual inspection and descriptive terms such as “bright” or “weak” fluorescence. Such observations are inherently qualitative and subjective. Quantitative fluorescence measurements would make these data more objective and comparable.
- We were unable to use our current instrument to quantify. Our main objective was to determine whether HcPro constructs localized to the nucleus and cytoplasm as the wild-type.
- The discrepancies between the PTGS and VIGS results are not clearly explained. It was mentioned that some PTGS mutants lost activity, while in VIGS they retained it, but the authors do not discuss why. A possible biological interpretation of this observation is missing.
- We have removed VIGS section
- Although the purpose of the study is to explore the relationship between SIE and silencing suppression, section 2.3.2 does not clearly show how the VIGS results relate to SIE.
- Same answers as point two above
- Compared to section 2.3.1 (PTGS), experiment 2.3.2 is described much more briefly, with fewer methodological details and limited interpretation.
- Same answers as point two above
- In section 2.3.1, HA-tag immunoblotting was performed to confirm HcPro expression, but this verification is missing for the VIGS experiment. It remains unclear whether all HcPro variants were expressed at comparable levels when delivered via the PVX vector.
- Same answers as point two above
- No quantitative correlation with the PTGS data is provided. The authors state that the results “generally coincide,” but no statistical comparison or correlation analysis is shown.
- Same answers as point two above
- The manuscript indicates that mutations (E419D, D419E, RVLN/IRK) affect VIGS efficiency, but does not discuss why these positions may be important, for instance, their potential structural or functional role in HcPro.
- Same answers as point two above
- The exact description of the PVX vector used in this work is not specified in the Methods section, which is important for reproducibility.
- Same answers as point two above
- Although the qRT-PCR data analysis is described in detail, the presentation of the results lacks transparency. The authors state that the data represent three biological replicates with three technical repeats; however, individual data points are not shown on the graphs, making it difficult to assess biological variability. It is strongly recommended to display individual replicates on the graphs and to include a supplementary table with raw or normalized ΔCt/ΔΔCt values to improve reproducibility and clarity.
- There was a lot of effort made in producing the graphs with the SE of means, without the specific experimental points, the graphs still show differences in values. While we acknowledge that adding the points is a way of presenting the data, we hope leaving the graphs as they are does take away from the conclusions
- The Y-axes in different panels use inconsistent scales, which complicates visual comparison between experiments. Standardizing axis scales across all panels would facilitate a more accurate interpretation of relative changes.
- There were very wide differences in the values of 2-ΔΔCt and it would really be impossible having the same scale; more importantly, each construct in each experiment was compared to an empty plasmid, not between different plasmids. Thus, empty plasmid/HcPro (on each side of the leaf blade) were individual experimental units throughout this work
Minor comments:
- The authors should consider adding information about the PVY strains used in the study to the abstract.
- There is a restriction on word count, we provided our previous publication on these strains
- The rationale for selecting these particular PVY strains should be clarified and justified in the manuscript.
- We added that these strains are found in North America
- The quality and resolution of the figures 6 and 7 should be improved to enhance data visibility.
- The image has been replaced.
Reviewer 3 Report
Comments and Suggestions for Authors
Dear Editor,
IJMS (ISSN 1422-0067)
I would like to express our sincere gratitude for selected me to review your journal manuscript. You can find my positive feedback and comments provided on your manuscript ID: ijms-3979136 as following:
- The references cited in the manuscript needs to be up-to-date, so add more newly related ones in the all manuscript sections.
- For the PVY strains, I recommend following References outlined in the DOI I have provided below on page no 1 - introduction section:
https://doi.org/10.1186/s41938-019-0165-1
Kindly include it in your citations after References [2, 3]
https://doi.org/10.1186/s43088-025-00686-y
Kindly include it in your citations after References [4]
- Please write a full name before the abbreviations (P1, HcPro, P3, 6K1, CI, 6K2, VPg, Nia, NIb, and CP) on the page no 1 – introduction section as the first time mentioned.
- Please write a full name before the abbreviations (P3N-PIPO) – (RdRp) on the page no 2 – introduction section as the first time mentioned
- The phenomenon of cross-protection mentioned on the page no 2 – introduction section, and the page no 12 – discusion section this phenomenon is recently renamed as systemic acquired resistance (SAR), so, please change it.
- The introduction provides not a clear statement of the problem and not enough the proposed approach or solution. Please provide it more efficient additions especially how the SIE acts for PVY inhibition.
- The references [60, 67] are in inappropriate self-citations by authors and needs to be change with their mentioned paragraphs in Introduction section.
- On the end of introduction, it is not clear whether authors want to explain their manuscript aim of work or mention the results specific of other literatures cited in reference [42]. So, I suggested to adding the clear work aims on this point.
- Please write a write the complete scientific name (Agrobacterium tumifaciens) on the page no 13 – Materials and methods section as the first time mentioned.
- Please insert the related reference for the freeze-thaw method on the page no 13 – Materials and methods section.
- Please insert the related reference for a chemiluminescence (ECL) method on the page no 14 – Materials and methods section.
- The author mentioned in subtitle Western blot analysis the page no 14 – Materials and methods section that the proteins separated in a 4-20% SDSPAGE gel. This is not accurate because it is very wide range of SDS-PAGE concentration and I think 12% is more suitable.
- Please write a full name before the abbreviations (YFP) on the page no 15 – Materials and methods section as the first time mentioned
- The authors must be insert new subtitle (Statistical analyses) in the end of Materials and methods section, with more details different analyses and deleted the replicative mentioned methods that cited wrong in the results section
- The section Materials and methods needs to rearrangement and moved 1. Plasmid construction and inoculations to be paste before subtitle 4.4. Western blot analysis.
- The figure 1 is not author`s private data, so it must be moved from the page no 3 – Results section and paste in Introduction section supported by suitable paragraph.
- The authors mentioned unsuitable paragraphs in Results section talks about the methods of (HcPro suppression of PTGS) and (HcPro suppression of VIGS). So, it must be moved in the same subtitles in the Materials and methods section.
- It should be added more details clears that authors added the value of SE or SD in figures 2, 3, 4 footnote.
- The authors should be adding the Confocal microscopy magnifications in figures 5, 6, 8 footnote.
- The authors should be adding the conclusion section.
- The authors should be adding the novelty section.
- The authors should be adding the list of abbreviations section.
Author Response
- The references cited in the manuscript needs to be up-to-date, so add more newly related ones in the all manuscript sections.
For the PVY strains, I recommend following References outlined in the DOI I have provided below on page no 1 - introduction section:
https://doi.org/10.1186/s41938-019-0165-1
Kindly include it in your citations after References [2, 3]
https://doi.org/10.1186/s43088-025-00686-y
Kindly include it in your citations after References [4]
- We added some of the authors that applied to the contect
- Please write a full name before the abbreviations (P1, HcPro, P3, 6K1, CI, 6K2, VPg, Nia, NIb, and CP) on the page no 1 – introduction section as the first time mentioned.
- This was corrected
- Please write a full name before the abbreviations (P3N-PIPO) – (RdRp) on the page no 2 – introduction section as the first time mentioned
- This was corrected
- The phenomenon of cross-protection mentioned on the page no 2 – introduction section, and the page no 12 – discusion section this phenomenon is recently renamed as systemic acquired resistance (SAR), so, please change it.
- This was corrected
- The introduction provides not a clear statement of the problem and not enough the proposed approach or solution. Please provide it more efficient additions especially how the SIE acts for PVY inhibition.
- This was corrected
- The references [60, 67] are in inappropriate self-citations by authors and needs to be change with their mentioned paragraphs in Introduction section.
- We routinely use this established same protocol in our lab
- On the end of introduction, it is not clear whether authors want to explain their manuscript aim of work or mention the results specific of other literatures cited in reference [42]. So, I suggested to adding the clear work aims on this point.
- This was corrected
- Please write a write the complete scientific name (Agrobacterium tumifaciens) on the page no 13 – Materials and methods section as the first time mentioned.
- This was corrected
- Please insert the related reference for the freeze-thaw method on the page no 13 – Materials and methods section.
- This was added
- Please insert the related reference for a chemiluminescence (ECL) method on the page no 14 – Materials and methods section.
- These details were provided
- The author mentioned in subtitle Western blot analysis the page no 14 – Materials and methods section that the proteins separated in a 4-20% SDSPAGE gel. This is not accurate because it is very wide range of SDS-PAGE concentration and I think 12% is more suitable.
- Yes indeed, the gel and source were added
- Please write a full name before the abbreviations (YFP) on the page no 15 – Materials and methods section as the first time mentioned
- This was corrected
- The authors must be insert new subtitle (Statistical analyses) in the end of Materials and methods section, with more details different analyses and deleted the replicative mentioned methods that cited wrong in the results section
- This was corrected
- The section Materials and methods needs to rearrangement and moved 1. Plasmid construction and inoculations to be paste before subtitle 4.4. Western blot analysis.
- This was corrected
- The figure 1 is not author`s private data, so it must be moved from the page no 3 – Results section and paste in Introduction section supported by suitable paragraph.
- This was corrected
- The authors mentioned unsuitable paragraphs in Results section talks about the methods of (HcPro suppression of PTGS) and (HcPro suppression of VIGS). So, it must be moved in the same subtitles in the Materials and methods section.
- VIGS was removed
- It should be added more details clears that authors added the value of SE or SD in figures 2, 3, 4 footnote.
- This was added
- The authors should be adding the Confocal microscopy magnifications in figures 5, 6, 8 footnote.
- This was added
- The authors should be adding the conclusion section.
- This was added
- The authors should be adding the novelty section.
- This was corrected
- The authors should be adding the list of abbreviations section.
- We tried to, but there were very few unknown to the readership.
Round 2
Reviewer 2 Report
Comments and Suggestions for Authors
The authors have addressed several of my previous comments, and in particular the removal of the VIGS section has resolved a number of serious methodological concerns. However, a few issues remain that, in my opinion, should still be clarified or adjusted in the manuscript.
I list them below.
1. Previously, I suggested showing individual biological replicates in the graphs and providing a supplementary table with raw or normalized ΔCt/ΔΔCt values. The authors declined, arguing that means ± SE are sufficient and that adding points is only one possible format.
While the current graphs do show mean differences, including individual data points and/or a supplementary table is best practice and strongly supports transparency and reproducibility. I therefore still recommend, at minimum, a supplementary table with raw Ct or normalized ΔCt/ΔΔCt values. If the journal treats this as optional, I defer to the editor, but consider it a valuable improvement for independent evaluation and data reuse.
2. I could not identify in the revised manuscript where the authors added a rationale for the selection of the specific PVY strains. I recommend briefly explaining why these strains are relevant, ideally in one or two clarifying sentences.
Author Response
We have made revisions as below:
While the current graphs do show mean differences, including individual data points and/or a supplementary table is best practice and strongly supports transparency and reproducibility. I therefore still recommend, at minimum, a supplementary table with raw Ct or normalized ΔCt/ΔΔCt values. If the journal treats this as optional, I defer to the editor, but consider it a valuable improvement for independent evaluation and data reuse.
- We have attached means of 2-ΔΔCT for these experiments.
I could not identify in the revised manuscript where the authors added a rationale for the selection of the specific PVY strains. I recommend briefly explaining why these strains are relevant, ideally in one or two clarifying sentences.
- We have included in the text that the four strains PVYNTN, but not PVYO, PVYN-Wi, and PVYN:O were chosen because these strains are common in North America and the study is a follow up to our recent study in Niraula et al. (2024).